# Theoretical analysis of deep neural networks for temporally dependent observations

**Mingliang Ma**
Department of Statistics
University of Florida
Gainesville, FL 32611
maminglian@ufl.edu

**Abolfazl Safikhani**
Department of Statistics
George Mason University
Fairfax, VA 22030
asafikha@gmu.edu

## Abstract

Deep neural networks are powerful tools to model observations over time with non-linear patterns. Despite the widespread use of neural networks in such settings, most theoretical developments of deep neural networks are under the assumption of independent observations, and theoretical results for temporally dependent observations are scarce. To bridge this gap, we study theoretical properties of deep neural networks on modeling non-linear time series data. Specifically, non-asymptotic bounds for prediction error of (sparse) feed-forward neural network with ReLU activation function is established under mixing-type assumptions. These assumptions are mild such that they include a wide range of time series models including auto-regressive models. Compared to independent observations, established convergence rates have additional logarithmic factors to compensate for additional complexity due to dependence among data points. The theoretical results are supported via various numerical simulation settings as well as an application to a macroeconomic data set.

## 1 Introduction

Neural networks have the ability to model highly complex relationship among data. If input data are observed data in past with future observations as response, neural networks can be utilized to perform time series forecasting. Examples of application of neural networks in forecasting include biotechnology [1], finance [24], health sciences [20], and business [29], just to name a selected few. Compared to more traditional time series forecasting methods such as ARIMA models [9], neural networks have the ability to detect highly non-linear trend and seasonality. In this work, we analyse the prediction error consistency of (deep) feed-forward neural networks to fit stationary (non-linear) time series models.

The property of the single hidden layer neural network is well studied in the past few decades. For example, [21] use single hidden layer neural network with a transformed cosine activation function to show that a sufficiently complex single hidden layer feed-forward network can approximate any member of a specific class of functions to any desired degree of accuracy. Such an approximation property for neural networks with sigmoidal activation function was also analyzed in [4, 5]. Further, [3] use monotonic homogeneous activation function (general version of ReLU activation function) and show that both the input dimension and number of hidden units have an effect on the convergence rate when using single layer neural networks.

There are many recent works which shed some light on the reasoning behind the good performance of multi-layer (or deep) neural networks. The performance is evaluated via computing mean squared predictive error which is also called the statistical risk. For example, [6] show that the statistical risk of a multi-layer neural network depends on the number of layers and the input dimension of

36th Conference on Neural Information Processing Systems (NeurIPS 2022).

each layer. The problem of applying deep neural network in high-dimensional settings is that the high-dimensional input vector in nonparametric regression leads to a slow convergence rate [8] while the complexity often scales exponentially with the depth or number of units per layer [7, 16, 23]. Further, convergence rate for prediction error is also related to the regression function and fast rate can be obtained in special classes of regression functions such as additive and/or composition functions [8, 17, 18]. To avoid the curse of dimensionality and achieving faster rates, [25] work under hierarchical composition assumption with a sparse neural network. It is shown that under the independence assumption over input vectors, the estimator utilizing a sparse network achieves nearly optimal convergence rates for prediction error. Finally, [26] use neural network as a classifier for temporally dependent observations which are based on Markov processes. We refer to [14] for an overview of deep learning methods.

Most of theoretical developments related to prediction error consistency of neural networks are under the assumption that either the input variables are independent or they are independent with the error (noise) term, or both. However, these assumptions are restrictive and may not hold in time series models. To bridge this gap, the main goal of this paper is to establish consistency rates for prediction error of deep feed-forward neural networks for temporally dependent observations. To that end, we focus on multivariate nonparametric regression model with bounded and composite regression functions and apply sparse neural networks with ReLU activation functions for estimation (see more details in Section 2). The modeling framework is similar to [25] while the independence assumption is relaxed. Specifically, we show that given temporally dependent observations, under certain mixing condition, the statistical risk coincides with the result of independent observations setting with an additional $\log^4(n)$ factor where $n$ is the sample size (Theorem 1). Moreover, utilizing the Wold decomposition, this result is extended to a general family of stationary time series models in which it is shown that the decay rate of AR($\infty$) representation coefficients plays an important role on the consistency rate for prediction error of neural networks (Theorem 2). These results give some insights on the effect of temporal dependence on the performance of neural networks by specifically quantifying the prediction error in such settings. Finally, the prediction performance of neural networks in time series settings is investigated empirically via several simulation scenarios and a real data application (Sections 4 and 5).

*Notation.* For two random variables $X$ and $Y$, $X \overset{D}{=} Y$ implies that $X$ and $Y$ have the same distribution. For a matrix $W$, $\|W\|_\infty := \max_{i,j} |W_{ij}|$, $\|W\|_0$ is the number of nonzero entries of $W$, and $\|W\|_1 := \sum_{i,j} |W_{ij}|$. For a vector $v$, $|v|_0$, $|v|_1$ and $|v|_\infty$ are defined by the same way. We write $\lfloor x \rceil$ for the smallest integer $\geq x$. For two sequences $(a_t)_{t \geq 1}$ and $(b_t)_{t \geq 1}$, we write $a_t \lesssim b_t$ if there exists a constant $c \geq 1$ such that $a_t \leq cb_t$ for all $t$. If both $a_t \lesssim b_t$ and $b_t \gtrsim a_t$, we write $a_t \asymp b_t$. Also, $a_t = o(b_t)$ implies $a_t/b_t \to 0$ as $t \to \infty$. For a multi-dimension random variable $\mathbf{X}$, $\mathbf{X} \sim N(\mu, \Sigma)$ implies that $X$ has multivariate Gaussian distribution with mean $\mu$ and covariance matrix $\Sigma$. For two functions $f, g$, we use $f \circ g(x)$ to denote $f(g(x))$. Also, $\|f\|_\infty := \sup_x |f(x)|$ is the sup-norm of $f$, and $(a)_+ = \max(a, 0)$ for $a \in \mathbb{R}$.

## 2 Setup

In this section, a brief presentation of feed-forward neural networks is provided in Section 2.1 followed by discussing the modeling framework under consideration in Section 2.2.

### 2.1 Background about multi-layer neural networks

Multi-layer neural network is composed of three parts: input layer, hidden layers and output layer. We denote the depth of a multi-layer neural network by $L$, which implies that there are $L + 1$ layers in total consisting of $L - 1$ hidden layers, one input layer and one output layer. We refer to the input layer as the 0-th layer and the output layer as the $L$-th layer. The multi-layer neural network can be written as

$$f(x) : \mathbb{R}^d \longrightarrow \mathbb{R} = W_L \sigma_{\mathbf{v}_L}(W_{L-1} \sigma_{\mathbf{v}_{L-1}}(\cdots W_1 \sigma_{\mathbf{v}_0}(W_0(x)))), \tag{1}$$

where $W_i$ is the matrix of weights between $(i-1)$-th and $i$-th layers of the network $(i = 1, \ldots, L)$ and $\sigma_{\mathbf{v}}$ is a modified ReLU activation function in each layer. Specifically, for the shift parameter

$\mathbf{v} = (v_1, \cdots, v_r) \in \mathbb{R}^r$, the activation function $\sigma_{\mathbf{v}} : \mathbb{R}^r \longrightarrow \mathbb{R}$ is defined as

$$\sigma_{\mathbf{v}} \begin{pmatrix} x_1 \\ \vdots \\ x_r \end{pmatrix} = \begin{pmatrix} (x_1 - v_1)_+ \\ \vdots \\ (x_r - v_r)_+ \end{pmatrix}.$$

Let $p_i$ denote the number of units in the $i$-th layer (note that $p_0 = d$ and $p_L = 1$). For a fully connected multi-layer neural network, the total number of parameters is $\sum_{i=0}^{L-1} p_i p_{i+1}$ that is defined as the size of a multi-layer neural network [10]. Similar to [25], the entries of all weight matrices $\{W_i\}_{i=1,\cdots,L}$ and shift parameters $\{\mathbf{v}_i\}_{i=1,\cdots,L}$ are assumed to be uniformly bounded. The sparsity level $s$ is defined as the number of all non-zero parameters in $\{W_i\}_{i=1,\cdots,L}$ and $\{\mathbf{v}_i\}_{i=1,\cdots,L}$. We also assume that the value of $|f|$ is bounded by some constant $F$. In summary, we focus on the collection of $s$-sparse multi-layer neural networks with bounded parameters which is denoted by $\mathcal{F}(L, p, s, F)$ and defined as

$$\mathcal{F}(L, \mathbf{p}, s, F) := \{f \in \mathcal{F}(L, \mathbf{p}) : \sum_{i=0}^{L} \|W_i\|_0 + |\mathbf{v}_i|_0 \leq s, \|f\|_\infty \leq F\},$$

$$\text{where } \mathcal{F}(L, \mathbf{p}) := \{f \text{ of form (1)} : \max_{i=0,1,\cdots,L} \|W_i\|_\infty \vee |\mathbf{v}_i|_\infty \leq 1\}.$$

This restriction of neural networks to the ones with sparse connections and bounded parameters is common on deep learning (see e.g. [25] and references therein) since neural networks are typically trained using certain penalization methods and dropouts.

## 2.2 Model

The modeling framework considered is similar to [25] while allowing for temporal dependence among observations. Let $(\epsilon_t)_{t \geq 1}$ be a sequence of independent random variables with $\mathbb{E}[\epsilon_t] = 0$. Let $\{\mathbf{X}_t\}_{t \geq 1}$ be a p-dimensional stationary process with $\mathbf{X}_t := (X_{t,1}, \cdots, X_{t,p})$. We assume $Y_t$ is generated as

$$Y_t = f_0(\mathbf{X}_t) + \epsilon_t, \tag{2}$$

with a measurable function $f_0 : \mathbb{R}^p \longrightarrow \mathbb{R}$. We assume that the regression function $f_0$ is a composition of several functions, specifically

$$f_0 = g_q \circ g_{q-1} \circ \cdots \circ g_1 \circ g_0, \tag{3}$$

with $g_i : [a_i, b_i]^{d_i} \longrightarrow [a_{i+1}, b_{i+1}]^{d_{i+1}}$, where $d_0 = p, d_{q+1} = 1$. Each $g_i$ has a $d_{i+1}$-dimensional vector output $g_i = (g_{i,1}, \cdots, g_{i,d_{i+1}})$. We assume that the multivariate function $g_{i,j}$ depends on at most $t_i$ variables while $t_i$ is far less than $d_i$, i.e. $t_i \ll d_i$. As mentioned in [25], for a $\beta$-smooth function $f_0$, the minimax estimation rate for the prediction error is $n^{-2\beta/(2\beta+d_0)}$. Since the dimensionality $d_0$ can be large in applications, the rate can be slow. To mitigate this issue, the sparse structure avoids the effect of input dimension on the convergence rate and improves the rate.

Let $T$ be a region in $\mathbb{R}^r$. Let $\beta$ and $L$ be two positive numbers. The Hölder class $\Sigma(\beta, L)$ is defined as the set of $\alpha = \lfloor \beta \rfloor$ times differentiable functions $f : T \longrightarrow \mathbb{R}^r$ whose derivative $\partial^\alpha f(\mathbf{x})$ satisfies

$$\frac{|\partial^\alpha f(\mathbf{x}) - \partial^\alpha f(\mathbf{y})|}{|\mathbf{x} - \mathbf{y}|_\infty^{\beta - \lfloor \beta \rfloor}} \leq L,$$

where we used the notation $\partial^\alpha = \partial^{\alpha_1} \cdots \partial^{\alpha_r}$ with $\alpha = (\alpha_1, \cdots, \alpha_r)$ and $|\alpha| := |\alpha|_1$. Further, we define the ball of $\beta$-Hölder functions with radius $K$ as

$$\mathcal{C}_r^\beta(D, K) = \{f : D \subset \mathbb{R}^r \longrightarrow \mathbb{R} :$$
$$\sum_{\alpha:|\alpha|<\beta} \|\partial^\alpha f\|_\infty + \sum_{\alpha:|\alpha|=\lfloor \beta \rfloor} \sup_{\mathbf{x}, \mathbf{y} \in D} \frac{|\partial^\alpha f(\mathbf{x}) - \partial^\alpha f(\mathbf{y})|}{|\mathbf{x} - \mathbf{y}|_\infty^{\beta - \lfloor \beta \rfloor}} \leq K\}.$$

We assume that all functions $g_{ij}$ for $i = 0, \cdots, q$ and $j = 1, \cdots, d_{i+1}$ belong to $\beta_i$-Hölder class $\mathcal{C}_{t_i}^{\beta_i}(D_{ij}, K)$. From the model (3), we know that $D_{ij} = [a_i, b_i]^{t_i}$. Hence, the class of $f_0$ we focus

belongs to

$$\mathcal{G}(q, \mathbf{d}, \mathbf{t}, \boldsymbol{\beta}, K) := \{ f_0 = g_q \circ \cdots \circ g_0 : g_i = (g_{ij})_j : [a_i, b_i]^{d_i} \longrightarrow [a_{i+1}, b_{i+1}]^{d_{i+1}},$$
$$g_{ij} \in \mathcal{C}_{t_i}^{\beta_i}([a_i, b_i]^{t_i}, K), \text{ for some } |a_i|, |b_i| \le K \}, \tag{4}$$

with $\mathbf{d} := (d_0, \cdots, d_{q+1}), \mathbf{t} := (t_0, \cdots, t_q)$ and $\boldsymbol{\beta} := (\beta_0, \cdots, \beta_q)$.

## 3  Main result

In this section, we present consistency results for prediction error of deep neural networks applied to model (2) followed by providing time series model examples satisfying the assumptions in Section 3.1. First, we need to introduce some notations and state the main assumptions under which the theoretical developments are established. For any estimator $\widehat{f}_n$ in the class $\mathcal{F}(L, p, s, F)$, we define (similar to [25]) $\Delta_n(\widehat{f}_n, f_0)$ to measure the difference between the expected empirical risk of $\widehat{f}_n$ and the global minimum over all networks in the class $\mathcal{F}(L, p, s, F)$ as

$$\Delta_n(\widehat{f}_n, f_0)$$
$$:= \mathbb{E}_{f_0} \left[ \frac{1}{n} \sum_{i=1}^n (Y_i - \widehat{f}_n(\mathbf{X}_i))^2 - \inf_{f \in \mathcal{F}(L, p, s, F)} \frac{1}{n} \sum_{i=1}^n (Y_i - f(\mathbf{X}_i))^2 \right].$$

The quantity $\Delta_n(\widehat{f}_n, f_0)$ plays a pivotal role in consistency properties of neural networks. The performance of $\widehat{f}_n$ is evaluated by the prediction error defined as

$$R(\widehat{f}_n, f_0) := \mathbb{E}_{f_0} \left[ (\widehat{f}_n(\mathbf{X}) - f_0(\mathbf{X}))^2 \right], \tag{5}$$

with $\mathbf{X} \stackrel{D}{=} \mathbf{X}_t$ and $\mathbf{X}$ is independent with $\{\mathbf{X}_t\}_{t \ge 0}$. Recall from Section 2.2, the regression function $f_0$ is in the class $\mathcal{G}(q, \mathbf{d}, \mathbf{t}, \boldsymbol{\beta}, K)$. To simplify notations, we define $\beta_i^* := \beta_i \prod_{\ell=i+1}^q (\beta_\ell \wedge 1), \phi_n := \max_{i=0,\cdots,q} n^{-\frac{2\beta_i^*}{2\beta_i^* + t_i}}$. The following assumptions are needed to present the first theorem.

**Assumption 1** *For all $i = 1, 2, \ldots$, $\mathbb{E}[\epsilon_i] = 0$, $\mathbb{E}[\epsilon_i^2] = \sigma^2$, and there exists some positive constant $c$ such that $\mathbb{E}[|\epsilon_i|^m] \le \sigma^2 m! c^{m-2}, m = 3, 4, \cdots$.*

**Assumption 2** *$\{\mathbf{X}_t\}$ is a strictly stationary and exponentially $\alpha$-mixing process. Recall that the $\alpha$-mixing coefficient of a stationary process $\{\mathbf{X}_t\}$ is define as*

$$\alpha(s) = \sup\{|\mathbb{P}(A \cap B) - \mathbb{P}(A)\mathbb{P}(B)| : -\infty < t < \infty, A \in \sigma(\mathbf{X}_t^-), B \in \sigma(\mathbf{X}_{t+s}^+)\},$$

*where $\mathbf{X}_t^-$ consists of the entire past of the process including $\mathbf{X}_t$, and $\mathbf{X}_t^+$ consists of its entire future. The process $\{\mathbf{X}_t\}$ is said to be exponentially $\alpha$-mixing if there exists some constant $\tilde{c} > 0$ such that $\log(\alpha(t)) \le -\tilde{c}t, t \ge 1$.*

**Assumption 3** *The error term $\epsilon_t$ is independent with $\{X_s, s \le t\}$.*

Assumption 1 is known as the Bernstein condition and implies that $\epsilon_t$ is a sub-exponential variable. This assumption is often used when we cannot assume $\epsilon_t$ is bounded. Assumption 2 is to control the dependence among input variables and holds for a wide range of time series models, see e.g. auto-regressive models in [12]. Assumption 3 controls dependence between input variables and error terms. A more stringent condition is to assume the whole error process $\{\epsilon_t\}_{t \ge 0}$ is independent with $\{X_s\}_{s \ge 0}$. However, this assumption is restrictive since it excludes auto-regressive model which is an important family of time series models. To avoid this, we only assume the current error term is independent of current and past input variables. Further, this assumption ensures that $\sum_t \epsilon_t f_0(\mathbf{X}_t)$ is a martingale which helps in verifying certain concentration inequalities needed in the proof of main results. All three assumptions are common in non-linear time series analysis, see e.g. [11]. Now, we are ready to state the main result.

**Theorem 1** *Consider the d-variate nonparametric regression model* (2) *for a composite regression function* (3) *in the class* $\mathcal{G}(q, \mathbf{d}, \mathbf{t}, \boldsymbol{\beta}, K)$. *Suppose Assumptions 1-3 hold. Let* $\widehat{f}_n$ *be an estimator taking values in the network class* $\mathcal{F}(L, (p_i)_{i=0,\cdots,L+1}, s, F)$ *satisfying (i)* $F \geq \max(K, 1)$; *(ii)* $\sum_{i=0}^{q} \log_2(4t_i \vee 4\beta_i)\log_2 n \leq L \lesssim n\phi_n$; *(iii)* $n\phi_n \lesssim \min_{i=1,\cdots,L} p_i$; *and (iv)* $s \asymp n\phi_n \log n$. *Then, there exist positive constants* $C, C'$ *depending only on* $q, \mathbf{d}, \mathbf{t}, \boldsymbol{\beta}, F$, *such that if* $\Delta_n(\widehat{f}, f_0) \leq C\phi_n L\log^6 n$, *then*

$$R(\widehat{f}_n, f_0) \leq C' \phi_n L\log^6 n, \tag{6}$$

*and if* $\Delta_n(\widehat{f}, f_0) \geq C\phi_n L\log^6 n$, *then*

$$\frac{1}{C'}\Delta_n(\widehat{f}, f_0) \leq R(\widehat{f}_n, f_0) \leq C' \Delta_n(\widehat{f}_n, f_0). \tag{7}$$

Based on Theorem 1, the prediction error defined in (5) is controlled by $\phi_n L\log^6 n$. From condition (ii), $L$ is at least of the order of $\log_2 n$. Thus, the rate in Theorem 1 becomes $\phi_n \log^7 n$. This rate for the case of independent observations is of order $\phi_n \log^3 n$ based on Theorem 1 in [25]. Compared to latter, our rate has an extra $\log^4 n$ factor, which compensates for additional complexity in verifying the prediction error consistency in the presence of temporal dependence among input variables. Further, note that for a fully connected neural network, the number of parameters is $\sum_{i=1}^{L-1} p_i p_{i+1} \gtrsim n^2 \phi_n^2 L$. We can see that this number is greater than the sparsity level $s$ which is of order $n\phi_n \log n$ based on condition (iv) in Theorem 1. Thus, it can be seen that condition (iv) restricts the neural network class to the ones with sparse architecture. In other words, at least $\sum_{i=1}^{L-1} p_i p_{i+1} - s$ units of the neural network is completely inactive.

## 3.1  Time series model examples

In this section, we introduce some (well-known) examples of time series models that satisfy the assumptions of Theorem 1. The first example is to let $\{\epsilon_t\}_{t\in\mathbb{Z}}$ and $\{\mathbf{X}_t\}_{t\in\mathbb{Z}}$ be two independent processes. This independence assumption implies that the input variables $\mathbf{X}_t$ are exogenous, thus Assumption 3 is automatically satisfied. Further, assume $\epsilon_t$ satisfy the moment conditions in Assumption 1 (for example, they have normal distribution) and $\mathbf{X}_t$ is a stationary and geometrically $\alpha$-mixing process. There are many examples of such processes including certain finite-order auto-regressive processes, see e.g. [15, 12]. The second example is to consider non-linear auto-regressive models, i.e. assume

$$X_t = g(X_{t-1}, \cdots, X_{t-d}) + \epsilon_t. \tag{8}$$

This is a special case of model (2) by setting $\mathbf{X}_t = (X_{t-d}, \cdots, X_{t-1})$ and $Y_t = X_t$. Assuming $\epsilon_t$'s are i.i.d. random variables with positive density in the real line and the boundedness of the function $g$, it can be shown that there exists a stationary solution to equation (8) while the solution is exponentially $\alpha$-mixing as well [2, 12]. In both examples, assumptions of Theorem 1 are satisfied, thus the results of this Theorem are applicable.

Now, we consider a more general time series model. Recall that by the well-known Wold representation, every purely nondeterministic stationary and zero-mean stochastic process $X_t$ can be expressed as $X_t = \sum_{i=0}^{\infty} a_i \epsilon_{t-i}$ where $\epsilon_t$ is a mean-zero white noise. Further, if $X_t$ has a non-vanishing spectral density and absolute summable auto-regressive coefficients, i.e. $\sum_{i=1}^{\infty} |\phi_i| < \infty$, it has the AR($\infty$) representation $X_t = \sum_{i=1}^{\infty} \phi_i X_{t-i} + \epsilon_t$ (see e.g. [27]). Motivated by this discussion, we consider a general family of times series models satisfying

$$X_t = \sum_{i=1}^{\infty} \phi_i X_{t-i} + \epsilon_t, \tag{9}$$

where $\epsilon_t$'s are i.i.d. errors. Independence among $\epsilon_t$'s is a strong assumption compared to only assuming that they are uncorrelated, but this is required for our theoretical analysis. The interesting fact about model (9) is that it is a linear model. However, since there are infinite covariates in this AR($\infty$) representation, training neural networks directly is impossible. The common solution is to

truncate the covariates and only consider the first few, i.e. approximate model (9) by an AR(d) model for some $d$. This approximation can successfully estimate second order structures of the original model (i.e. spectral density or auto-correlation function) if $d$ is selected carefully and under certain assumptions on the AR coefficients $\phi_i$'s (see e.g. [27]). Therefore, we follow this path and fit a neural network to the $d$-dimensional input variables $(X_{t-1}, \ldots, X_{t-d})$ with a proper choice of $d$ while keeping in mind that the true regression function is in fact $f_0(\mathbf{X}_t) = \sum_{i=1}^{\infty} \phi_i X_{t-i}$. To establish the prediction error consistency of neural networks on truncated input variables, we need two additional assumptions.

**Assumption 4** *There exists $\alpha > 0, M > 0$ such that $\sum_{i=1}^{\infty}(1+i)^{\alpha}|\phi_i| \leq M < \infty$.*

**Assumption 5** *For some constant $K > 0$, $|X_t| \leq K$ for all $t \geq 0$.*

Assumption 4 controls the decay rate of the AR($\infty$) coefficients in the true model and $\alpha$ can be treated as a decreasing rate of $\phi_i$'s. This assumption is needed to compensate for approximating a general time series of form (9) with a finite lag AR process. Further, it plays an important role in restricting the first derivative of $f_0$, which corresponds to the $\beta_i$-smoothness assumption on $g_{ij}$ in (4). Note that Assumption 4 is satisfied if the spectral density function is strictly positive and continuous, and the auto-covariance function of $X_t$ has some bounded property [19]. Moreover, since the model is linear (i.e. the regression function is unbounded), Assumption 5 becomes necessary to make $f_0(\mathbf{X}_t)$ bounded, a property needed for Theorem 1 as well.

**Theorem 2** *Consider model (9) with $f_0(\mathbf{X}_t) = \sum_{i=1}^{\infty} \phi_i X_{t-i}$. Let $\widehat{f}_n$ be an estimator taking values in the network class $\mathcal{F}(L, (p_i)_{i=0,\cdots,L+1}, s, F)$ satisfying (i) $F \geq KM$; (ii) $L \geq 4$; (iii) $s \asymp Ld$, and (iv) $d \lesssim \min_{i=1,\cdots,L} p_i$. Assume that $d \asymp n^{\frac{1}{\alpha+1}}$. Under the Assumptions 1-5, there exist positive constants $C, C'$ such that if $\Delta_n(\widehat{f}, f_0) \leq C n^{-\frac{\alpha}{\alpha+1}} L\log^5 n$, then*

$$R(\widehat{f}_n, f_0) \leq C' n^{-\frac{\alpha}{\alpha+1}} L\log^5 n, \tag{10}$$

*and if $\Delta_n(\widehat{f}, f_0) > C n^{-\frac{\alpha}{\alpha+1}} L\log^5 n$, then*

$$\frac{1}{C'} \Delta_n(\widehat{f}, f_0) \leq R(\widehat{f}_n, f_0) \leq C' \Delta_n(\widehat{f}, f_0). \tag{11}$$

Based on Theorem 2, the best convergence rate of $R(\widehat{f}_n, f_0)$ for model (9) is $n^{-\frac{\alpha}{\alpha+1}}\log^5 n$. Compared with the convergence rate in Theorem 1, we can see that the convergence rate in model (9) depends on the decreasing rate of coefficients $\phi_i$'s instead of the smoothness of regression function. Also the logarithmic factor changes from $\log^7 n$ to $\log^5 n$. Such a subtle decrease is due to the fact that since the truncated model is linear, a shallower and sparser neural network can be used in the proof of Theorem 2 (as seen from conditions (ii) and (iii) in the statement of Theorem 2). Also, note that for a simple linear AR(d) model where $\phi_j = 0$ for $j > d$, Assumption 4 is satisfied for any large $\alpha$. Thus, in this case, the best convergence rate of prediction error becomes $n^{-1}\log^5 n$.

**Remark 1** *To fit the neural network to a truncated model (9), shallow neural networks are sufficient. In fact, $L = 4$ is enough based on Theorem 2. This is because model (9) has a simple linear structure compared to composition functions (3).*

**Remark 2** *For a general selection of input vector $d$, the result of Theorem 2 becomes $R(\widehat{f}, f_0) \leq C\frac{1}{d^{\alpha}} + C'\frac{dL\log^5 n}{n} + 4\Delta_n(\widehat{f}, f_0)$. With the choice of $d \asymp n^{\frac{1}{\alpha+1}}$, we balance the first two terms and establish a proper upper bound for $R(\widehat{f}_n, f_0)$. Thus, this selection can be regarded as the "optimal selection" of lag $d$ when applying neural networks for estimation in model (9).*

## 4 Simulation experiments

In this section, we conduct two simulation settings to illustrate the performance of neural networks applied to temporally dependent data (see also Section C in the supplementary materials for numerical comparisons between feed-forward neural networks and LSTM). In the first simulation in Section 4.1, we aim to compare the convergence rate in dependent observations setting with the rate in independent

observations setting. The convergence rate is explained as how fast the mean squared predictive error decreases as the sample size grows. In the second simulation (Section 4.2), motivated by time series model examples introduced in Section 3.1, we use neural networks to fit linear and non-linear auto-regressive models and compare the performance with the result of linear regression method (least squares method). All simulations are repeated 200 times.

To train the neural network, we split the data into three parts: training set $\mathcal{T}_1$, validation set $\mathcal{T}_2$, and testing set $\mathcal{T}_3$. We set $\mathcal{T}_1 = \{1, \cdots, n/2\}$, $\mathcal{T}_2 = \{n/2, \cdots, 3n/4\}$, and $\mathcal{T}_3 = \{3n/4, \cdots, n\}$. We penalize parameters of weight matrix of the neural network and use mean square error as our loss. To be more specific, the loss function is defined as

$$\text{loss} := \sum_{t \in \mathcal{T}_1} (Y_t - \widehat{Y}_t)^2/(n/2) + \lambda \sum_{i=1}^{L} \|W_i\|_1, \tag{12}$$

where $\lambda$ is the sparsity tuning parameter. The value of $\lambda$ does not have significant effect on simulation results. In this section, we set $\lambda = 0.1$. We apply gradient descent method to update parameters of neural network and stop the iteration when the neural network gives the minimum mean square error for $\mathcal{T}_2$. The prediction error $R(\widehat{f}_n, f_0)$ defined in (5) is empirically estimated by $\widehat{R} = \sum_{t \in \mathcal{T}_3} (f_0(\mathbf{X}_t) - \widehat{Y}_t)^2/(n/4)$. We use $\widehat{R}$ to evaluate the performance of the trained neural network.

## 4.1 Dependent vs. independent observations

In this experiment, we consider a nonlinear additive model $f_0(\mathbf{X}_t) = 2 \sum_{i=1}^{4} \cos(X_{t,i})$. Thus, $Y_t$ is from the following model

$$Y_t = 2 \sum_{i=1}^{4} \cos(X_{t,i}) + \eta_t, \quad t = 1, 2, \cdots, n, \tag{13}$$

with $\eta_t$'s as independent standard normal random variables, and $\mathbf{X}_t := (X_{t,1}, \cdots, X_{t,4})$ is generated from an AR(4) model. Specifically,

$$\mathbf{X}_t = \begin{bmatrix} 0 & \rho & 0 & 0 \\ 0 & 0 & \rho & 0 \\ 0 & 0 & 0 & \rho \\ 0 & 0 & 0 & 0 \end{bmatrix} \mathbf{X}_{t-1} + \boldsymbol{\epsilon}_t, \quad \boldsymbol{\epsilon}_t \sim N(\mathbf{0}, \mathbf{I}_4),$$

where $\rho$ takes two value, $\rho = 0.2$, 0.6. The sample size is $n = 100, 400, 1600, 6400$. A three layer neural network is selected to fit model (13) while ReLU is used as the activation function. The network requires a 4- dimensional input vector and has 20 units in each hidden layer. To make a fair comparison with the performance of neural network in independent observations setting, we still use model (13) but with independent observations, i.e. we generate $\mathbf{X}_1', \cdots, \mathbf{X}_n' \sim N(\mathbf{0}, \boldsymbol{\Sigma})$ i.i.d. and $\boldsymbol{\Sigma}$ derives from $\mathbf{X}_1' \overset{D}{=} \mathbf{X}_1$.

Figure 1 illustrates $\log\left(\widehat{R}\right)$ against various sample sizes for both temporally dependent case and independent case by running 200 replications for each. As can be seen, the prediction performance of neural network improves as sample size increases. Further, we observe that the prediction error for the neural network with temporally dependent data and independent one are similar. This may imply that the additional logarithmic terms appearing in Theorem 1 might be an artifact of the proof.

## 4.2 Auto-regressive examples

In this experiment, we test the performance of the neural network and compare its result with simple linear regression (least squares method) for several linear and non-linear auto-regressive models. Specifically, we consider the following four models: (1) $X_t = 0.6X_{t-1} + \epsilon_t$; (2) $X_t = 0.6X_{t-1} - 0.4X_{t-2} + 0.2X_{t-3} + \epsilon_t$; (3) $X_t = 0.5\sqrt{|X_{t-1}|} + \epsilon_t$; (4) $X_t = 0.5|X_{t-1}| + \epsilon_t$. The error term is generated as $\epsilon_1, \cdots, \epsilon_n \sim N(0, 1)$ i.i.d. in all models. The sample size is $n = 100, 400, 1600, 6400$.

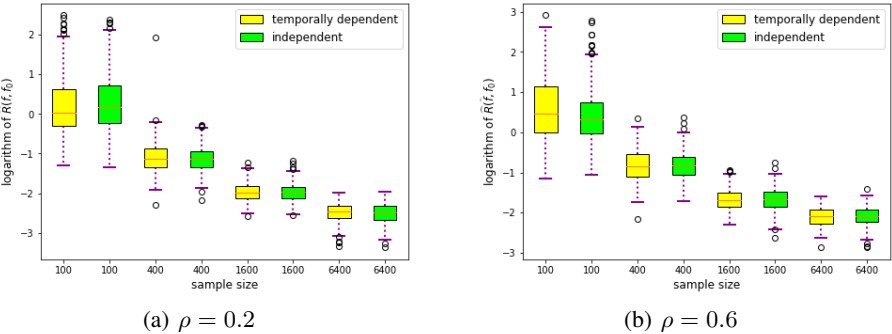

(a) $\rho = 0.2$              (b) $\rho = 0.6$

Figure 1: Box plots of logarithm of the mean square error on the testing set as a function of sample size.

We again use a four layer neural network with 20 units in each hidden layer. This time, we have no prior knowledge on the input dimension of the neural network, since lags of these four time series models are unknown. To determine the input dimension (lag of time series), we apply linear regression with AIC criterion. The AIC is defined as AIC $= n\log(SSE) + 2d$, in which $d$ is the input dimension and $SSE$ is the summation of squared errors.

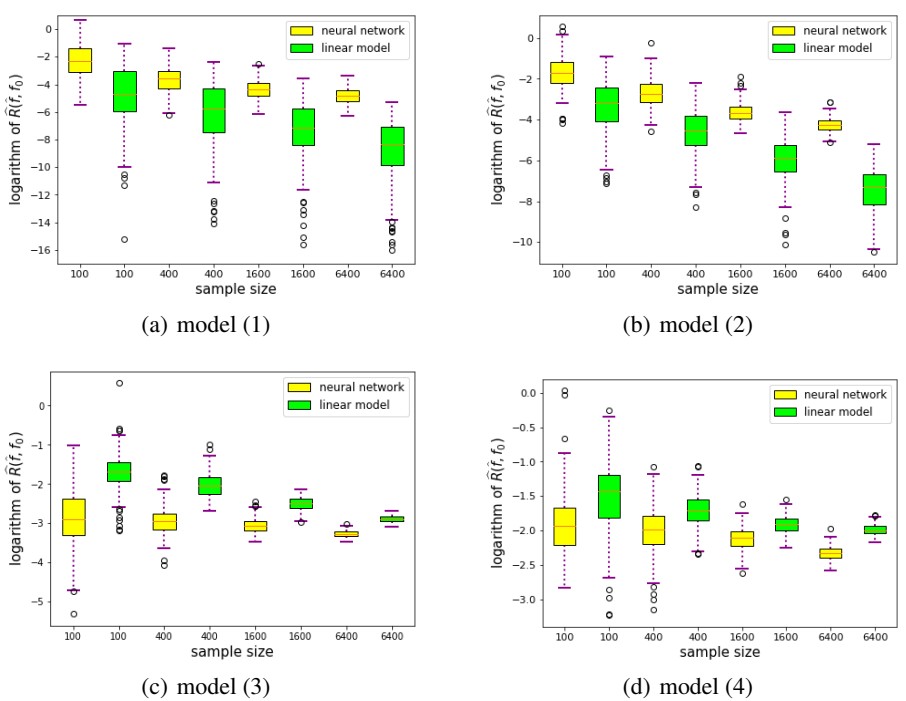

(a) model (1)           (b) model (2)

(c) model (3)           (d) model (4)

Figure 2: Box plots of logarithm of the mean square error on the testing set as a function of sample size.

Figure 2 displays the logarithm of $\widehat{R}(\widehat{f}, f_0)$ as sample size increases. Linear regression method has a fast convergence rate in AR(1) and AR(3) models, i.e. models (1) and (2). For non-linear autoregressive models (models (3) and (4)), the convergence rate of linear regression method becomes slow while neural network outperforms the linear method in these cases. In summary, neural network has the advantage to deal with estimation in both linear and non-linear auto-regressive models.

## 5 Real data analysis

In this section, we apply neural network on a macroeconomic data[1] to predict monthly inflation rate. The data consist of 132 monthly macroeconomic variables from January 1960 to December 2011, a total of 624 time periods. The inflation rate is measured by the percentage changes of the Consumer Price Index (CPI). To be more specific, we are interested in forecasting

$$\pi_t = 1200 \times \log\left(\frac{\text{CPI}_{t+1}}{\text{CPI}_t}\right). \tag{14}$$

Similar to [22, 28], we use the AR(4) model as the benchmark model, i.e. $\pi_{t+1} = \alpha_0 + \sum_{i=0}^{3} \alpha_i \pi_{t-i}$. Thus, the overall model can be written as

$$\pi_{t+1} = f(\pi_t.\pi_{t-1}, \pi_{t-2}, \pi_{t-3}, \mathbf{x}_t) + v_{t+1}, \tag{15}$$

where $\mathbf{x}_t$ consists of other 131 predictors in the data set along with three of their lags. Therefore, $\mathbf{x}_t$ is a 524-dimensional vector. We consider one step ahead forecasts computed in a rolling window scheme with 453 observations. Since there are some missing values in the first four years, we start from 1964 to fit model and predict the inflation rate. We use the observations between May 1964 and Jan 2002 to fit (15) and predict the monthly inflation rate starting at time Feb 2002.

We apply the feature screening method Sure Independent Screening (SIS) proposed in [13] to reduce the input dimension and select only the top $\gamma\%$ important variables from $\mathbf{x}_t$ for $\gamma = 0, 5, 7.5, 10$. The case of $\gamma = 0$ implies that we do not select any covariates from $\mathbf{x}_t$ and only use four lags of $\pi_t$ as predictors. Then we use the selected covariates and four lags of $\pi_t$ as predictors or the input vector of our neural network. The neural network is selected to have ReLU as the activation function and 100 units in each hidden layers. Two different depths of neural network are selected, one with 3 layers (1 hidden layers) and the other with 6 layers (4 hidden layers). To train the neural network, we use dropout method for hidden layers and gradient descent to update parameters. In each rolling window, 453 observations are divided into two parts: the first 300 observations is treated as training set while the last 153 observations is treated as validation set. The loss function is again (12) with $\lambda = 0.1$. We stop training when MSE over the validation set reaches its minimum. The performance is evaluated by MSE of one step ahead forecast. We also show the performance of linear regression (i.e. least squares method) with the same input as neural network.

The results are summarized in Table 1 while Figure 3 in the supplementary materials plots the predicted values against the truth. As seen from Table 1, neural network has a better performance compared to linear regression method. In other words, the overall prediction error of neural network with three layers is small compared to the ones for linear regression (except for $\gamma = 7.5$). Specifically, in the case of $\gamma = 10$ (i.e. including 52 covariates from vector $\mathbf{x}_t$), the performance of neural network remains satisfactory while linear regression method suffers from large input vector dimension. Finally, note that three layers seems to be enough for this data as increasing the number of layers does not reduce the prediction error.

|  | neural network (3 layers) | neural network (6 layers) | linear regression |
|---|---|---|---|
| $\gamma\% = 0\%$ | **15.71** | 17.44 | 18.05* |
| $\gamma\% = 5\%$ | **15.65** | 18.37 | 17.05 |
| $\gamma\% = 7.5\%$ | 16.40 | 19.67 | **15.42** |
| $\gamma\% = 10\%$ | **16.51** | 20.70 | 172.49 |

Table 1: Prediction error of inflation rate, i.e. the mean square of the forecasting error. Since the result of neural network depends on initialization, we averaged results across 10 replicates. The entry with asterisk corresponds to the benchmark model (i.e. AR(4) model).

## 6 Conclusion

Considering nonparametric regression model with the regression function belonging to a specific family of bounded composite functions, we analyzed the performance of deep feed-forward neural

---

[1]link : https://www.sydneyludvigson.com/data-and-appendixes

networks with ReLU activation function on estimating such functions. Consistency of prediction error is established under mild conditions on the input data which can include temporal dependence among observations. Interestingly, the consistency rate matches the one for independent data with additional logarithmic factors with respect to sample size. This result is applicable to a wide range of linear and non-linear time series models including finite lag non-linear auto-regressive models. Then, the result is extended to include a general family of stationary time series models utilizing the Wold decomposition while the consistency rate depends on the decay rate of $AR(\infty)$ representation coefficients. Relaxing some assumptions in the theoretical analysis including the mixing condition and boundedness of observations in the case of general time series models are interesting future directions. Another limitation of the work is considering only the ReLU as the activation function while extending the results to a more general family of activation functions is a fruitful research direction.

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
