# OpenReview forum: "Theoretical analysis of deep neural networks for temporally dependent observations"
_NeurIPS.cc/2022/Conference — NeurIPS 2022 Accept_

### Official Review · Reviewer_FsYW · 2022-07-01

**Rating:** 5
**Confidence:** 4
**Soundness:** 2 fair
**Presentation:** 3 good
**Contribution:** 3 good

**Summary:**

In this paper, the authors derive the expected generalization bounds on (sparse) feed-forward neural networks with ReLU activation functions. This model was considered in Johannes Schmidt-Hieber (2020) [reference 25]. However, different from Johannes Schmidt-Hieber's model, the authors assume that the dataset is non-i.i.d. More specifically, the authors assume that the observation sequence is a strictly stationary and exponentially $\alpha$-mixing process, which includes time series models such as auto-regressive models. The results show that a multiplicative term of order $\log^3 n$ is added to the generalization bound in [25] (for i.i.d. dataset) which explains for the additional difficulty of prediction caused by the correlation among samples in the dataset. Some numerical results and an application to a macroeconomic dataset are given to support the theoretical results.

The main idea of this paper (including proofs) is originated from [25]. However, there are some changes in Beinstein inequalities to account for the dependence among samples. More specifically, the Beinstein inequality in [25] for i.i.d. process (dataset) is replaced by two versions in this paper: (1) Beinstein inequality for $\alpha$-mixing process and (cf. line  68) (2) Beinstein inequality for martingale difference sequence (cf. line 26) thanks to the assumptions (1)-(3) in this paper and some algebraic manipulations. Lemma 3 is the key contribution in this paper.

**Questions:**

In general, all the results in this paper are well proved and look correct. There are some typos and unclear parts. Please see the following list for some typos or concerning problems.

1. Please explain in details why we obtain the expression $(2k-1)var(Y_{n_0})$ in the line 23? It should be $var(Y_{n_0})+2\sum_{i=1}^{k-1} |cov(Y_{n_0},Y_{n_i})|$.
2. The sum in Lemma 2 should be $2\sum_{n\geq i>0}|cov(Y_{n_0},Y_{n_i})|$ since the fact in line 22 only holds for $i\leq n$. Please check whether these limit affects your results.
3.  In Lemma 3, $\mathcal{N}(\delta,\mathcal{F},\lVert \cdot\rVert_{\infty})$ should be defined ($\delta$-cover of $\mathcal{F}$ under $\|\cdot\|_{\infty}$ norm.)
4. In the proof of (II) in Lemma 3 proof, please change $\hat{f}$ to $\tilde{f}$.
5. The $\sigma$-algebra $\mathcal{F_t}$ in line 89 should be defined as $\mathcal{F_t}:=\sigma((\mathbf{X_i}):i\leq t)$ instead of $\mathcal{F_t}:=\sigma((\mathbf{X_i},\epsilon_i):i\leq t)$. Otherwise, the inequalities such as between line 92 and 93 are hard to explain except when you put an additional asssumption that $(\epsilon_i: i \in [n])$ are independent in Assumption 1. Besides, you also need to assume that $\mathbb{E}[\epsilon_i]=0$ in Assumption 1 such that some expressions in the proof hold (for example, $\mathbb{E}[\epsilon_i f_0(\mathbf{X_i})]=0$ in line 83).


**Limitations:**

This is a theoretical work. The authors already finished the checklist as well as stated the limitations of theorems and results. Some additional constraints as mentioned above should be added.

**Strengths And Weaknesses:**

This work can be considered as a combination of well-known techniques with some new elements in using concentration inequalities as mentioned in the summary part. The strong point is that these results are new and have not appeared in the research literature (to my knowledge). In addition, as mentioned, the authors had some new contributions in making use of some new Beinstein inequalities from the technical viewpoint.  However, most of the arguments resemble [25], and the changes look not novel since they are much dependent on assumptions 1-3, which is the main weak point of this paper.

The deep neural (worst-case) generalization networks with temporally dependent observation, including the Auto-regressive model, was considered in some existing results (for example, https://arxiv.org/abs/2201.11059). Please cite these related results.

---

> ### Author Response · Authors · 2022-08-02
> **response to comments by Reviewer FsYW**
>
> We thank the reviewer for great questions/comments, please see our responses below:
>
> (Q1): Please explain in details why we obtain the expression $(2k-1)\mathrm{Var}(Y_{n0})$ in the line 23? It should be $\mathrm{Var}(Y_{n0}) + 2\sum_{i=1}^{k-1}|\mathrm{cov}(Y_{n0},Y_{ni})|$.
>
> (A1): Both expressions are correct. Ii seems that the potential confusion comes from the definition of $Y_{ni}$. Note that $Y_{ni}$ is defined as $a_nX_{i}$, where $a_n$ has no relationship with $X_i$ and can be treated as a constant when we fix $n$. The transformed time series $Y_{ni}$ is not a truncated sequence, which means that $i$ is allowed to take the value from $0$ all the way to the infinity. According to the definition of $Y_{n0}$, $|\mathrm{cov}(Y_{n0},Y_{ni})| = a_n^2 |\mathrm{cov}(X_{0},X_{i})|\leq a_n^2 \mathrm{var}(X_0) = \mathrm{var}(Y_{n0})$. We can conclude that $\mathrm{Var}(Y_{n0}) + 2\sum_{i=1}^{k-1}|\mathrm{cov}(Y_{n0},Y_{ni})| \leq (2k-1)\mathrm{Var}(Y_{n0})$ accordingly.
>
> (Q2): The sum in Lemma 2 should be $2\sum_{n\geq i>0}|\mathrm{cov}(Y_{n0},Y_{ni})| $ since the fact in line 22 only holds for $i\leq n$. Please check whether these limit affects your results.
>
> (A2): As mentioned in (A1), $Y_{ni}$ is not a truncated sequence which means that $i$ can take the value from 1 to the infinity. The inequality $Y_{ni} := a_n f(X_i) \leq n^{\alpha}M $ holds for all $i\in \mathbb{Z}$ (no restriction like $i\leq n$).
>
> (Q3): In Lemma 3, $\mathcal{N}(\delta,\mathcal{F},\Vert \cdot\Vert_{\infty})$ should be defined ($\delta$-cover of $\mathcal{F}$ under $|\cdot|_{\infty}$ norm).
>
> (A3): The definition is given in line 25 (supplement file).
>
> (Q4): In the proof of (II) in Lemma 3 proof, please change $\widehat{f}$ to $\tilde{f}$.
>
> (A4): Thanks for pointing out this typo. We will fix this in the revision.
>
> (Q5): The $\sigma$-algebra in line 89 should be defined as $\mathcal{F}_t := \sigma((\mathbf{X_i}),i\leq t)$ instead of $\mathcal{F}_t := \sigma((\mathbf{X_i},\epsilon_i),i\leq t)$. Otherwise, the inequalities such as between line 92 and 93 are hard to explain except when you put an additional assumption that $(\epsilon_i, i\leq n)$ are independent in Assumption 1. Besides, you also need to assume that $\mathbb{E}[\epsilon_i]=0$ in Assumption 1 such that some expressions in the proof hold (for example, $\mathbb{E}[\epsilon_i f_0(\mathbf{X_i})] =0$  in line 83).
>
>
> (A5): Thanks very much for pointing this out. Yes, the $\sigma$-algebra in line 89 should be defined as $\mathcal{F}_t := \sigma((\mathbf{X_i}),i\leq t)$. We will fix this in the revision. For the assumption $\mathbb{E}\epsilon_t = 0 $, we have mentioned it in the first paragraph in Section 2.2. To avoid confusion, we will add this to assumption 1 in the revision.
>
> (Q6): Most of the arguments resemble [25], and the changes look not novel since they are much dependent on assumptions 1-3, which is the main weak point of this paper.
>
>
> (A6): Assumptions used in this paper are common in time series literature, see e.g. [1,2]. We agree with the reviewer that relaxing some of the assumptions can further strengthen our paper. For example, the mixing assumption may be relaxed to weak dependence. We will add a discussion about this in the revision.
>
>
> (Q7): Citing additional papers in the literature.
>
> (A7): Thanks for pointing to an interesting paper in the literature, we will cite it in the revision.
>
> [1] Davis, R. A., \& Nielsen, M. S. (2020). Modeling of time series using random forests: Theoretical developments. Electronic Journal of Statistics, 14(2), 3644-3671.
>
> [2] Wong, K. C., Li, Z., \& Tewari, A. (2020). Lasso guarantees for $\beta $-mixing heavy-tailed time series. The Annals of Statistics, 48(2), 1124-1142.

---

### Official Review · Reviewer_D8dK · 2022-07-11

**Rating:** 4
**Confidence:** 3
**Soundness:** 2 fair
**Presentation:** 2 fair
**Contribution:** 2 fair

**Summary:**

This paper provides a theoretical analysis of nonparametric regression for dependent observations using deep neural networks. Non-asymptotic bounds for prediction errors are established. Numerical studies and empirical applications confirms the validity of the theoretical results.

**Questions:**

To find the DNN estimator, how to perform optimization under the constraints $\|f\|_{\infty}\leq F$, $\max\|W_i\|_0+\|v_i\|_0\leq s$ given tuning parameters $F$ and $s$.

**Strengths And Weaknesses:**

Weakness: Even though the authors provide upper bounds for the DNN estimator, it seems that the result is an extension from I.I.D observations to dependent data.

---

> ### Author Response · Authors · 2022-08-02
> **response to comments by Reviewer D8dK**
>
> We thank the reviewer for great questions/comments, please see our responses below:
>
> (Q1): To find the DNN estimator, how to perform optimization under the constraints $|f|_{\infty} \leq F$, $\mathrm{max}|W_i|_0+ |v_i|_0 \leq s$, given tuning parameters $F$ and $s$.
>
> (A1): How to train DNN with constrains in practice is not the main focus of our work. Our aim is to show that under certain conditions, there exists one DNN estimator with close to optimal rate of convergence. In our experiments, the sparsity constrain is achieved by dropout and regularization method while the boundedness constraint is only for theoretical purposes.
>
> (Q2): Even though the authors provide upper bounds for the DNN estimator, it seems that the result is an extension from I.I.D observations to dependent data.
>
> (A2): Theoretical results for DNN with dependent observations are quite limited in the literature. We are not aware of theoretical analysis of DNN in the presence of temporal dependence. One reason is that it is hard to control for the dependence among observations when dealing with such a non-linear structure. The goal of this paper is to bridge this gap. Specifically, when applying concentration inequalities to dependent data and dealing with martingale difference terms, we need to care about this dependence. Nonetheless, our result is quite close to the optimal rate when data are independent. In summary, our paper provides a theoretical support for the convergence property of the neural network in certain temporally dependent cases.

---

### Official Review · Reviewer_KBnt · 2022-07-11

**Rating:** 6
**Confidence:** 2
**Soundness:** 3 good
**Presentation:** 3 good
**Contribution:** 3 good

**Summary:**

This paper establishes some non-asymptotic bounds on the prediction error of using feedforward neural network to learn models when the data are timeseries with temporal dependence. These bounds are established under assumptions on uniform bounds on the network parameters, with the ReLU activation function, and assumptions of the model (which is rich enough to contain finite AR process).

Performance of this structure is then tested on a few experiments and compared with regression.

**Questions:**

The comparison of temporally dependent and independent data suggests there is not the large difference expected from the theorem. And it is postulated that the additional logarithmic term may be an artifact. This implies the bounds obtained are not tight, how might this result be tightened further?

For the linear AR examples, wouldn't the linear regression be a rich enough class to learn the true model (or am I misinterpreting what would be the predictor)? If so, why does the approximation error remain high for model 1 and 2 in Figure 2?

**Limitations:**

The authors briefly discuss in the conclusion the limitation of their work in terms of the assumptions they have placed, i.e. they are studying a rather specific family of problems bounded observations and ReLU activation function for the neural network.

Also, as mentioned above the bounds do not appear tight. How might this result be tightened further in the restricted case examined in this paper.

Although not discussed, this paper assumes that the output dimension is 1, how might the analysis change for greater dimensions?

**Strengths And Weaknesses:**

The paper is fairly well written and adds to the literature by giving theoretical bounds for the case of temporal dependent data in the simple case of a feedforward neural network. The code for the experiments are included for full reproducibility.

However, it appears the bound is not tight, so the experiments did not show as big a difference between temporally dependent and independent cases as expected from the theory.

It is interesting to be able to obtain optimality bounds. However, as in machine learning a lot of the results are empirical, I would be interested in seeing how the model performs on prediction tasks against something like LSTM/GRU which has structure to capture time-dependence. I would expect that LSTM would perform better in which case, the applicability/significance of the results of this paper would be lower given the alternative, better neural network structure.

---

> ### Author Response · Authors · 2022-08-02
> **response to comments by Reviewer KBnt**
>
>
> We thank the reviewer for great questions/comments, please see our responses below:
>
> (Q1): The comparison of temporally dependent and independent data suggests there is not the large difference expected from the theorem. This implies the bounds obtained are not tight, how might this result be tightened further?
>
> (A1): We believe that tighter bounds can be derived with simpler models, like linear AR(p) model and/or m-dependent processes. Given the current modeling framework in our paper, it is hard to remove the extra logarithmic terms in the final result. The logarithmic factor always appears when we apply Bernstein inequality in temporally dependent cases unless much tighter Bernstein inequality is available. As for the optimal bound, [1] gives the minimax rate of convergence for independent observations. His result states that as long as the regression function $f\in \mathcal{G}(q, d, t, \beta, K)$ (defined in line 111), the optimal rate for nonparamteric method is $\phi_n$. Compared with our rate ($\phi_n$ in only multiplied by a logarithmic factor), our result is quite close to the optimal rate. If one is interested in getting a sharp decrease in rate of convergence, it seems that considering other class of model functions may be another solution.
>
>
> (Q2): For the linear AR examples, wouldn't the linear regression be a rich enough class to learn the true model (or am I misinterpreting what would be the predictor)? If so, why does the approximation error remain high for model 1 and 2 in Figure 2?
>
> (A2): Yes, since finite AR model is a linear model, linear regression is the perfect approach to fit the model. As discussed in [2], linear regression can consistently estimate transition matrices (i.e. model parameters) in high-dimensional vector auto-regressive models. The rate is similar to the case of independent observations while the constants are different. The additional constant terms are related to temporal dependence (eigenvalues of spectral density matrices) existing among observations. These additional constant terms may be high and result in high approximation error in finite samples when linear regression method is applied.
>
>
> (Q3) It is interesting to be able to obtain optimality bounds. However, as in machine learning a lot of the results are empirical, I would be interested in seeing how the model performs on prediction tasks against something like LSTM/GRU which has structure to capture time-dependence. I would expect that LSTM would perform better in which case, the applicability/significance of the results of this paper would be lower given the alternative, better neural network structure.
>
> (A3) LSTM/GRU and DNN have similar performance when applied to short memory processes. The main advantage of LSTM is that it can use the long term memory with relatively small sample size compared with DNN. Specifically, LSTM performs better when the information from the distant past has a significant effect on the current state. We applied LSTM to the four simulation studies in the paper and it seems the performance of LSTM and DNN based on mean square prediction error are similar; see e.g. the table below re model 1. We will add all details about this comparison and a discussion about this in the revision.
>
>
> (Q4) This paper assumes that the output dimension is 1, how might the analysis change for greater dimensions?
>
>
> (A4) Neural network with multivariate output is a natural extension. In the case of fixed output dimension, we can still use square error as the loss function, then the proof still works and the rate of convergence doesn't change due to separability of loss function among model dimensions. To be more specific, we could use concentration inequalities and martingale inequalities for each dimension separately. If the output dimension is fixed (which excludes the high-dimension case), we can combine those inequalities to reach the final result.
>
> logarithm of the mean square error on the testing set (Model 1):
>
> |                  | quantile      | 25      | 50      | 75      |
> | ----------- | ----------- | ----------- | ----------- | ----------- |
> | n = 100     | LSTM       |  -0.481      |  -0.913       |  -1.791       |
> |                  | DNN         |  -1.501      |  -2.136       |  -3.182       |
> | n = 200     | LSTM       |  -0.713      |  -1.504       | -2.827        |
> |                  | DNN         |  -1.894      |  -2.822       |  -3.613       |
> | n = 400     | LSTM       |  -0.925      |  -2.610       |  -3.362       |
> |                  | DNN         |  -2.936      |  -3.364       |  -4.079       |
>
>
>
>
>
> [1] Schmidt-Hieber, J. (2020). Nonparametric regression using deep neural networks with ReLU activation function. The Annals of Statistics, 48(4), 1875-1897.
>
> [2] Basu, S., \& Michailidis, G. (2015). Regularized estimation in sparse high-dimensional time series models. The Annals of Statistics, 43(4), 1535-1567.

---

> > ### Comment · Reviewer_KBnt · 2022-08-09
> > **Response**
> >
> > Thank you for the author(s) for the response.
> >
> > With regards to "We applied LSTM to the four simulation studies in the paper and it seems the performance of LSTM and DNN based on mean square prediction error are similar; see e.g. the table below re model 1", this is a bit unclear to me, as the table seems to suggest much lower errors for LSTM than DNN, which reinforces my point about the applicability of the work.

---

> > > ### Author Response · Authors · 2022-08-09
> > > **response**
> > >
> > > We thank the reviewer for reading our response.
> > >
> > > In regards to the new comment, looking at table, the median of log(MSE) for the DNN method in all sample sizes are actually less than the LSTM method. For example, when n = 100, LSTM reaches -0.913 while DNN reaches -2.136 which shows smaller prediction error for the DNN method. Looking at other quantiles of log(MSE), we see that DNN reduces the prediction error compared to LSTM while this reduction may not be (statistically) significant.

---

> > > > ### Comment · Reviewer_KBnt · 2022-08-09
> > > > **Response**
> > > >
> > > > Ah yes, I see that I misread the table the first time as I was expecting only mean square error from the earlier comment and did not parse the negative signs (to account for the logs). Thanks for pointing that out.
> > > >
> > > > Can you provide some intuition on why we observe such a difference? Is this some arbitrary or optimised structure? And why is the n=200 and n=400 case for DNN the same?

---

> > > > > ### Author Response · Authors · 2022-08-10
> > > > > **response**
> > > > >
> > > > > The difference in performances may be a finite sample issue which would vanish for diverging n. Also, it may come from the form of regression model. For AR(p) linear model, the last p observations contain all the information to make one-step prediction. While for AR(p) nonlinear model, Wold decomposition tells us that this nonlinear model can be represented as an infinity summation of all the past observations. It implies that there is still some information we can get from the distant past. Therefore, LSTM performs slightly better than DNN in some nonlinear cases, while worse than DNN in linear cases. Note that still their performances are similar (i.e. not statistically significantly different from each other) in short memory processes. As mentioned before, the main advantage of LSTM is when dealing with processes with long memory which may violate the assumptions of our theoretical results (see e.g. assumption 4 on how fast the AR coefficients need to decay).
> > > > >
> > > > > We selected the input dimension of LSTM and DNN by AIC. The hidden layer is set to be 10. Also, there was a typo in presenting the results for n=400 in the previous response. We have changed the previous table and added a new table to represent the result for a nonlinear AR(p) model. As seen from this table, DNN has better performance in some cases while LSTM performs better in some other cases.
> > > > >
> > > > > logarithm of the mean square error on the testing set (Model 4):
> > > > >
> > > > > |                  | quantile      | 25      | 50      | 75      |
> > > > > | ----------- | ----------- | ----------- | ----------- | ----------- |
> > > > > | n = 100     | LSTM       |   -2.291  |  -2.847  |  -3.247 |
> > > > > |                  | DNN         |   -2.414  |  -2.909  |  -3.489 |
> > > > > | n = 200     | LSTM       |   -3.033  |  -3.341  |  -3.590 |
> > > > > |                  | DNN         |   -2.814  |  -3.198  |  -3.481 |
> > > > > | n = 400     | LSTM       |   -3.329  |  -3.594  |  -3.879 |
> > > > > |                  | DNN         |   -2.823  |  -3.116  |  -3.472 |

---

### Meta-Review · Area_Chair_t3R3 · 2022-08-26

**Recommendation:** Accept
**Confidence:** Less certain

**Metareview:**

The reviewers consensus is that this manuscript is a near the borderline between acceptance and rejection due in large part to the results being somewhat natural extensions of existing results from [25] with the appropriate extension of the bounds as pointed out by reviewer FsYW.  While this limits the overall excitement about the results, it is an important extension that would be of interest to many readers as time-series data is far more realistic in many applications than is i.i.d. data.  There is a useful discussion between reviewer KBnt and the author about some numerical experiments.  The author is encouraged to consider if is possible to expand upon the numerical experiments in the manuscript to highlight as much as possible the difference achieve between the dependent entries in the time-series and those in the i.i.d. setting.

**Award:**

No

---

### Decision · Program_Chairs · 2022-09-14

Accept